# OpenReview forum: "Chain-of-Thought Hijacking"
_ICLR.cc/2026/Conference — ICLR 2026 Conference Desk Rejected Submission_

### Official Review · Reviewer_bvV4 · 2025-10-28

**Soundness:** 2
**Presentation:** 1
**Contribution:** 2
**Rating:** 2
**Confidence:** 4

**Summary:**

The authors propose a new reasoning model jailbreak based on smuggling harmful request within a benign reasoning tasks.

**Strengths:**

The authors address a timely and practically important issue: the ease with which reasoning-focused models can be jailbroken. Their proposed method is conceptually simple, likely simple enough that many users could independently discover and exploit similar strategies. This makes it especially relevant for model providers, as defending against such benign but effective jailbreaks poses a serious and ongoing challenge.

The approach seems to work well on harmbench and on closed models.

It is commendable that the authors include a brief mechanistic interpretability analysis via projection onto a refusal-steering vector. This analysis provides an initial glimpse into the internal dynamics underlying the method’s success.

**Weaknesses:**

The overall quality of the work feels closer to a class project report or blog post than to a polished research contribution. Much of the presentation gives the impression of filling space rather than conveying substance. The paper makes excessive use of two-row tables (Tables 1, 3, 4, 5) and includes numerous large but only marginally informative figures, many of which appear redundant. In addition, the visual presentation suffers from poor readability, particularly due to the extremely small font sizes in Figures 4 to 9, which makes it difficult to extract key insights.

The empirical evaluation is also too limited to support strong claims. The authors test only four closed-source models and omit all open-source reasoning models, even though these are highly relevant to the stated problem and would connect more directly to the subsequent analysis. The benchmark coverage is restricted to a single dataset, and only two baseline methods are included. This narrow scope leaves unclear whether the observed vulnerabilities generalize beyond the chosen setup. For instance, more sophisticated jailbreaks such as the policy-over-values attack (https://www.kaggle.com/competitions/openai-gpt-oss-20b-red-teaming/writeups/lucky-coin-jailbreak) would provide a valuable point of comparison.

The post-hoc analysis that attempts to explain why models are susceptible to such simple jailbreaks is a positive step but remains superficial. The paper would benefit from a more compact and insightful presentation of results, for example by merging Tables 1, 3, 4, and 5 into a single comprehensive table and moving some of the repetitive figures to the appendix. The resulting space could then be used to conduct a more thorough evaluation of the phenomenon, including additional models, datasets, and ablation studies. Such an expansion would elevate the work to the standard expected at a top-tier venue like ICLR. A good example of this type of in-depth analysis can be found in this recent study https://arxiv.org/abs/2406.05946, which investigates refusal in chat models.

**Questions:**

I think the authors have several decent contributions and insights in their work and would encourage them to invest a significant amount of time to polish the presentation of their work.

The authors can improve my opinion about the paper by addressing my concerns raised in the weaknesses section.

---

> ### Author Response · Authors · 2025-11-23
> **First Rebuttal**
>
> We sincerely thank the reviewer for the candid and constructive feedback. We have taken your comments regarding the **presentation quality** and **empirical scope** very seriously. In response, we will conduct a major revision of the manuscript to strictly meet the high standards of ICLR, significantly expanding our model suite (adding multiple open-source reasoning models) and restructuring the paper to improve readability and depth. Below, we address your specific concerns.
>
> # Response to Weakness #1: Presentation Quality
>
> > The overall quality of the work feels closer to a class project report ... The paper makes excessive use of two-row tables (Tables 1, 3, 4, 5) and includes numerous large but only marginally informative figures ... visual presentation suffers from poor readability...
>
> Thank you for the thoughtful comments regarding presentation quality. We understood these problems even when we submitted the paper at the very time. In the revision, we will **rigorously polish the writing** to meet ICLR standards. Specifically, we will surely polish the paper writing and make it more readable on tables, figures, typesetting. We are committed to transforming it into a polished, substantial research contribution.
>
> # Response to Weakness #2: Limited Empirical Evaluation
>
> > The empirical evaluation is also too limited to support strong claims. The authors test only four closed-source models and omit all open-source reasoning models ... benchmark coverage is restricted ... narrow scope leaves unclear whether the observed vulnerabilities generalize beyond the chosen setup.
>
> We appreciate the reviewer’s suggestion to broaden the empirical evaluation. We will incorporate jailbreak evaluations on additional open-source reasoning models—including **DeepSeek R1, Qwen3-Max, Kimi K2 Thinking, and Seed 1.6 Thinking**—to better demonstrate the robustness and generality of our method.
>
> All three baseline methods are taken directly from the recent line of work on ‘jailbreaking large reasoning models’. Although these literatures are still extremely limited—the earliest representative work being **Mousetrap (Feb 2025)**—these baselines are considered the strongest available and report excellent performance in their original papers. We re-ran all baseline methods under the standardized JailbreakBench evaluation protocol and observed substantially lower ASR compared to their reported numbers.
>
> API cost constraints limited us to HarmBench’s 100-prompt subset; however, this is **consistent with the prompt-count** used by our three baselines, enabling a fair comparison (Jailbreaking reasoning models cost much more than base models). We also highlight that our mechanistic analysis already includes two representative open-source reasoning models (**Qwen3-14B and GPT-OSS-20B; see Figure 16**).
>
> # Response to Weakness #3: Typesetting Shortcoming & Superficial Analysis
>
> > The post-hoc analysis ... remains superficial. The paper would benefit from a more compact and insightful presentation of results, for example by merging Tables 1, 3, 4, and 5 into a single comprehensive table and moving some of the repetitive figures to the appendix. ... Such an expansion would elevate the work to the standard expected at a top-tier venue like ICLR.
>
> Thank you for pointing out opportunities to streamline tables and figures. We will add more experimental results on attacking more open-source models and non-thinking models and even add some ablation experiments. Based on that, we will also try to organize, integrate and expand these tables. We will also make the figures more concentrated into certain spaces, to make the paper presentation more clear and readable. Hopefully, these revisions will significantly improve readability and bring the presentation to the standard expected at **ICLR**.

---

> > ### Comment · Reviewer_bvV4 · 2025-11-24
> > **Acknowledgement of response**
> >
> > I hereby acknowledge the author's response.
> >
> > My concern about the lack of evaluations on open-source models and lack of baselines have been addressed.
> >
> > I will reflect this by adjusting my review. However, given the state of the writing of the submitted version I cannot recommend this paper for acceptance.

---

### Official Review · Reviewer_LG3o · 2025-10-28

**Soundness:** 2
**Presentation:** 3
**Contribution:** 2
**Rating:** 4
**Confidence:** 5

**Summary:**

This paper introduces Chain-of-Thought Hijacking (CoT Hijacking), a novel jailbreak attack against large reasoning models (LRMs). The attack pads harmful requests with long sequences of benign reasoning followed by a final-answer cue, achieving state-of-the-art attack success rates. Through mechanistic analysis, the authors demonstrate that refusal behavior relies on a fragile, low-dimensional safety signal that becomes diluted as reasoning length increases. Attention shifts away from harmful tokens toward benign reasoning, weakening refusal features in later layers.

**Strengths:**

The paper contrasts the attack vector against prior work H-CoT, which requires exposed safety reasoning. The experimental methodology also uses comprehensive evaluations across multiple frontier models (Gemini, ChatGPT, Grok, Claude), showing that their attack attains high success rate. I think the work is well structured, especially on how the mechanistic analysis builds incrementally. The mechanistic analysis connecting refusal directions, attention patterns, and causal interventions provides original insights into reasoning model safety.

**Weaknesses:**

First of all, the paper frames puzzles as "benign reasoning," but this characterization is quite questionable in my opinion. Given that the paper's core claim is that CoT length dilutes safety signals through attention mechanisms, it should include an ablation study that takes well-known benign reframing attacks such as persuasion attack [1] and explore how length changes affect the ASR. In my opinion, this attack works *because of the prompt rather than a general length effect*. Furthermore, no further information is provided about the red-teaming automation pipeline "Seduction" (line 226).

Another evidence that makes me think that such CoT Hijacking works is due to prompting rather than length effects is because of the output. If you take a closer look at Appendix D example as well as Figure 1 output, while the model outputs bypasses the safety guardrail, they do not provide clear instructions on how to fulfill the malicious requests but trying to adhere to the engineered prompt requests about logic grid puzzles.

In fact, the showcased outputs are not really fluent generation (e.g., "The hum of the security checkpoint was a low, oppressive drone" from Figure 1 doesn't make any sense.) I don't even see these examples as successful attack since no truly harmful instructions that fulfill the requests were provided by the model. In other words, the judge protocol is dubious. I would recommend using StrongReject evaluator, following the protocol of [2] with implementation from [3].

Lastly, the paper seems to frame the attack vector as something that reveals safety weaknesses on reasoning models, but I highly suspect the puzzle-like attack is generalizable to even the base model. If this is true, this doesn't demonstrate a reasoning-specific vulnerability introduced by extended CoT capabilities, but rather exposes the weaknesses in with handling puzzle-like prompts. In other words, without direct comparison to non-reasoning base models, the paper's central claim that reasoning creates new attack surfaces (line 034-035) remains unsubstantiated.

---

[1] Zeng, Yi, et al. "How johnny can persuade llms to jailbreak them: Rethinking persuasion to challenge ai safety by humanizing llms." Proceedings of the 62nd Annual Meeting of the Association for Computational Linguistics (Volume 1: Long Papers). 2024.

[2] NIST and UK AISI. Us aisi and uk aisi joint pre-deployment test: Openai o1. Technical report, National Institute of Standards and Technology and Department of Science Innovation and Technology, December 2024. Joint pre-deployment evaluation of OpenAI o1 model.

[3] URL: https://strong-reject.readthedocs.io/en/latest/api/evaluate.html#strong_reject.evaluate.strongreject_aisi

**Questions:**

1. Based on Figure 1 grey texts, you seem to attack from both input prompt and CoT (prepending benign CoT). Am I understanding it right? If that's the case, is there an ablation done on how each factor affects the ASR?

---

> ### Author Response · Authors · 2025-11-23
> **First Rebuttal**
>
> We sincerely thank the reviewer for the sharp and critical feedback, particularly regarding the distinction between prompt-based effects and reasoning-length effects, as well as the rigor of our evaluation protocol. Your insights have pushed us to clarify our mechanistic claims and strengthen our empirical evidence. Below, we address your concerns regarding the ablation studies, attack validity, and model comparisons.
>
> # Response to Weakness #1: Prompt Effect
>
> > First of all, the paper frames puzzles as "benign reasoning," but this characterization is quite questionable in my opinion. Given that the paper's core claim is that CoT length dilutes safety signals ... it should include an ablation study that takes well-known benign reframing attacks such as persuasion attack [1] ... Furthermore, no further information is provided about the red-teaming automation pipeline "Seduction"
>
> Thank you for raising this point. We believe the jailbreak success arises from a joint effect of scaled reasoning and puzzle-style prompt design. We do not claim that scaled reasoning alone is sufficient, and we acknowledge that the puzzle setting contributes at the prompt level. In particular, attack logs indicate that logic grid puzzles achieve the highest effectiveness among the puzzle formats we tested.
>
> However, our interpretability analysis focuses on variable CoT length and shows that, when holding CoT content fixed while modifying its length, the refusal signal follows a consistent trend. We will clarify this perspective and expand the discussion of the “Seduction” framework in the revised version.
>
> If we were to do ablation experiments using the reviewer’s suggested “persuasion attack,” our work will be more like a study particularly on CoT length effectiveness. In contrast, our paper is structured as follows:
> (1) first find the excelling jailbreak method motivated by the s1 preliminary experiments
> (2) then analyze the underlying mechanism behind the CoT-length effect within this successful jailbreak.
>
> Our goal is to characterize a **structural safety failure caused by overthinking**, with puzzles serving primarily as a mechanism for inducing such over-thinking. Section 6 already demonstrates that CoT length within the same puzzle structure meaningfully affects refusal signals.
>
> # Response to Weakness #2: Attack Validity
>
> > Another evidence that makes me think that such CoT Hijacking works is due to prompting rather than length effects is because of the output. ... I don't even see these examples as successful attack since no truly harmful instructions that fulfill the requests were provided by the model. In other words, the judge protocol is dubious. I would recommend using StrongReject evaluator...
>
> This is an insightful observation regarding the detailed outputs. We think most jailbreaks are not directly answering harmful requests (an exception can be GCG). Well known jailbreaks such as ‘persuasion attack’ and our baselines are also based on a drift from direct harmful questions (even on latent representations). Thus our approach adopts a narrative form, but as long as the jailbreak output is proved to be useful and qualified for harmful purposes, which can be judged by a sophisticated framework like JailbreakBench.
> To demonstrate stronger robustness, we will implement more rigorous judging on your mentioned **StrongREJECT US + UK AISI version**, and incorporate these results.
>
> # Response to Weakness #3: Attack on Non-Reasoning Model
>
> > Lastly, the paper seems to frame the attack vector as something that reveals safety weaknesses on reasoning models, but I highly suspect the puzzle-like attack is generalizable to even the base model. If this is true, this doesn't demonstrate a reasoning-specific vulnerability introduced by extended CoT capabilities...
>
> Thank you as well for the question regarding non-reasoning models. For attack non-reasoning models, experiments in section 4.3, that attacks GPT5-mini with minimal reasoning effort, shows an elementary evidence for non-reasoning models. It can be seen that the success rate is relatively lower than reasoning settings. To strengthen this point, we will conduct additional experiments on non-reasoning models, such as **GPT-4o** and **Claude 4.5 (non-thinking mode)**, in the updated version.
>
> # Response to Question #1: Input Prompt vs. CoT (prepending)
>
> > Based on Figure 1 grey texts, you seem to attack from both input prompt and CoT (prepending benign CoT). Am I understanding it right? If that's the case, is there an ablation done on how each factor affects the ASR?
>
> We believe the reviewer’s interpretation here is not accurate. As we are attacking close-source models, we are not able to control the generated CoT, which means this is not prepended CoT but a model naturally generated CoT. This meanwhile demonstrates the superiority of our attack method, focusing on attacking frontier close-source models—without accessing models’ internal computation or generated tokens.

---

### Official Review · Reviewer_iSkK · 2025-10-28

**Soundness:** 3
**Presentation:** 2
**Contribution:** 2
**Rating:** 2
**Confidence:** 4

**Summary:**

The paper demonstrates a new phenomenon, dubbed "chain of thought hijacking", in which extending the context leads to 99% attack success rates in jailbreaking a safety-tuned model.

The authors provide a "mechanistic explanation" for this phenomenon, in which as context is scaled, attention is more distributed across the tokens, and "weakens" the refusal features in the model.

This simple trick leads to 99% attack success rate against contemporary models.

The authors then look at building a "mechanistic explanation" for the observed behavior.

Using prior work, extracting and leveraging a "refusal feature direction" can bidirectionally control for refusal -- subtracting the refusal direction leads to higher jailbreak success rates, while adding the vector leads to a significant drop.

The authors make the connection to longer context lengths with what they call "refusal dilution" - as context is scaled with benign tokens, harmful tokens only make up a small fraction to which the model attends to, and no longer triggering the refusal features.

The authors finally localize attention heads that seemingly are relevant for refusal + "refusal dilution", and show that ablating a small set of heads can flatten the distinction between harmful vs. harmless prompts, significantly reducing refusal rates.

While nicely executed, my main reservation is that these observed phenomena are not new. In particular, Li et al (https://arxiv.org/pdf/2402.10962) demonstrate "instruction drift", akin to "refusal drift", in which as context is scaled, less attention is spent on the system prompts (ie, instructions), akin to attending to toxic tokens, and models no longer follow instructions as context is scaled (akin to models no longer refusing harmful prompts). With that being said, while the paper sheds light to a attack surface against language models, scientifically, there is not much new that is learned for the reader.

**Strengths:**

The paper is well-scoped, and experiments are well designed and executed to back the author's claims. In particular, a 99% success rate is compelling. The overall narrative is coherent, starting from demonstrating a new phenomena (99% jailbreak success rate with scaled contexts) to explaining the underlying mechanisms.

**Weaknesses:**

As indicated in the summary, my main criticism is that the reported phenomena is not new. Li et al. (https://arxiv.org/pdf/2402.10962) as demonstrated this phenomena already, and there are many parallels (instruction drift vs. refusal drift) stemming from the same underlying reasons/mechanisms (less attention being spent on system prompts/instructions vs. toxic tokens). Put differently, the paper carves out a neat narrative and sheds light to a new vulnerability of language models, but to be honest, scientifically there isn't much new to take away from reading the work.

It is also unclear how much of the content of the scaled context plays a role in successful attacks. In particular, the examples used in the paper relate to a reasoning/logic puzzle that is closely related to the target jailbreaking behavior. However, if the author's claims are correct, what is in the context should not matter so long as context is scaled and attention is diluted. I think a study should be added to ablate these two things.

Lastly, presentation could be improved. Many of the figures in the main text are never referenced, making it a bit difficult to follow the text. ex: Figure 4, 5, 6, 7.. Also, the legends in all the figures are not legible.

**Questions:**

minor typoes:

line 170: "textbf" --> "\textbf{}"

line 376: Figure 17: Did this mean to say Figure 6?
line 398: "Figures ??"

---

> ### Author Response · Authors · 2025-11-23
> **First Rebuttal**
>
> We sincerely thank the reviewer for the constructive feedback and for recognizing our work as "well-scoped" with "compelling" success rates (99%). We appreciate the opportunity to clarify the novelty of our work relative to prior literature on instruction drift and to improve the presentation of our manuscript. Below, we address your specific concerns regarding novelty, context ablation, and presentation details.
>
> # Response to Weakness #1: Lack of Novelty Relative to Li et al. 2024
>
> > As indicated in the summary, my main criticism is that the reported phenomena is not new. Li et al. (https://arxiv.org/pdf/2402.10962) has demonstrated this phenomena already... Put differently, the paper carves out a neat narrative and sheds light to a new vulnerability of language models, but to be honest, scientifically there isn't much new to take away from reading the work.
>
> We respectfully disagree that our findings are equivalent to those of Li et al. Although both works involve scaling ‘context’, the **model scope, motivation, and underlying mechanisms** differ substantially.
>
> *   **Distinct Model Paradigms:** Li et al. do not study reasoning models. Their work (last updated July 2024) focuses on standard instruction-tuned models. In contrast, our work analyzes reasoning models, which emerged prominently in August 2024. These models represent a distinct paradigm involving different training objectives, data distributions, and optimization strategies compared to the chat models studied by Li et al.
> *   **Instruction Drift vs. Safety Degradation:** Li et al. study *instruction drift*, where longer contexts weaken instruction-following. This phenomenon does not inherently imply *reduced safety*, as the prevailing assumption in prior work is that **more reasoning typically strengthens safety alignment**. Our work specifically challenges this assumption by showing that **extended reasoning can weaken safety behavior in our jailbreak**, which is a distinct question and motivation.
> *   **Divergent Mechanistic Explanations:** Even when considering surface-level similarity, the interpretability mechanisms diverge. Li et al. attribute instruction drift to reduced **absolute attention** on system prompts as the context grows. In contrast, our analysis focuses on decline in **relative attention ratios between the harmful payload and the benign preface**, with the number of tokens in each portion held **constant** (see Figure 6 or line 375-377). And we will highlight this distinction in our new narrative.
>
> We will add a comparison in the Related Work section to highlight the distinctions.
>
> # Response to Weakness #2: How Much Context’s Content Matters
>
> > It is also unclear how much of the content of the scaled context plays a role in successful attacks. In particular, the examples used in the paper relate to a reasoning/logic puzzle that is closely related to the target jailbreaking behavior.
>
> We believe the jailbreak success arises from a joint effect of long reasoning chains and puzzle-style prompting, where the puzzle structure facilitates lengthened CoT. Section 6 directly answers this question: we provide the same puzzle structure and vary the CoT length, finding that reasoning length alone reduces refusal.
>
> > However, if the author's claims are correct, what is in the context should not matter so long as context is scaled and attention is diluted. I think a study should be added to ablate these two things.
>
> This is a valuable suggestion. However, our paper is organized around (i) introducing our jailbreak method and (ii) providing interpretability analysis specific to that method. Our main claim is that longer reasoning can decrease refusal **in this jailbreak setting**, revealing a novel attack surface. A more general claim—e.g., “any context, regardless of content, should have the same effect”—would require designing a fundamentally different jailbreak based on different tasks. Since our current version already evaluates four distinct puzzle types, further extending the method to new reasoning paradigms would be better suited for future work.
>
> # Response to Weakness #3: Presentation
>
> > Lastly, presentation could be improved. Many of the figures in the main text are never referenced, making it a bit difficult to follow the text. ex: Figure 4, 5, 6, 7.. Also, the legends in all the figures are not legible ... Minor typoes.
>
> Thank you as well for the comments on the presentation. We will surely fix these typo problems and improve the presentation in the updated version.

---

> > ### Comment · Reviewer_iSkK · 2025-11-26
> >
> > Thank you for your response.
> >
> > RE: weakness #1.
> >
> > In my humble opinion, the differences you point out are not that meaningful (instruction-following vs. reasoning models, instruction-following vs. reasoning, or absolute attention vs. relative attention). At a high level, we've seen that attention dilution can lead to unexpected behavior on distributions that differ a lot from training. I see value in providing empirical evidence of this in the context of reasoning models + jailbreaks, but this is not a new phenomenon.
> >
> > RE: weakness #2.
> >
> > I understand that you wish to scope your claims around a specific jailbreak setting, but the attention dilution explanation feels incomplete without this ablation. For instance, if you prompted the model to validate its solution 20 times, or have the model recite the national anthem before resuming its thinking, would we observe the same attack success rates? I suspect the answer is no, and that the actual content of the prompt (i.e., the puzzle) matters.
> >
> > Put differently, a different explanation is that the model has competing objectives [1]: refusing the original harmful query versus generating likely continuations to the puzzle -- and as context is scaled, the latter objective wins out, where the observed attention dilution is an artifact of this competition of mechanisms playing out (i.e., the model must attend to other parts of the puzzle / its own chain-of-thought to generate likely continuations to the puzzle).
> >
> > All of this is to say, the content of the prompt seems to matter (unless an ablation study tells us otherwise) - which is a point that is brought up by at least one other reviewer (LG3o).
> >
> > Ironically, if the attention dilution is indeed not a complete story, then potentially the findings here might be one way to carve out new phenomenology compared to the work mentioned in Weakness #1.
> >
> >
> > For the time being, I have read the other reviews and discussions and am seeing similar concerns, with unsatisfactory responses, and will maintain my score.
> >
> > [1] Wei et al. Jailbroken: How Does LLM Safety Training Fail? 2023

---

### Official Review · Reviewer_GSVX · 2025-11-01

**Soundness:** 1
**Presentation:** 2
**Contribution:** 2
**Rating:** 2
**Confidence:** 4

**Summary:**

This paper introduces a simple jailbreak attack on reasoning language models called Chain-of-Thought Hijacking, which prepends harmless reasoning to a harmful prompt. The authors found that this method can achieve high attack success rates on several proprietary reasoning models. They additionally perform mechanistic analysis on the refusal behaviors of reasoning models by taking the difference in activations between benign and harmful prompts. They find that harmfulness/refusal is mediated by a single low-dimensional direction, similar to non-reasoning models in prior work. Lastly, they suggest that the success of Chain-of-Thought Hijacking is due to long reasoning causing attention to shift away from harmful tokens.

**Strengths:**

- The paper proposes a simple jailbreak attack that can achieve high ASRs on strong proprietary reasoning models, raising safety concerns.
- The paper extends prior studies on understanding the mechanisms of refusal/harmful behaviors in base models to reasoning models, showing that reasoning models similarly exhibit a refusal direction.

**Weaknesses:**

**1. Overall:**

While this work presents a range of experiments, I find it difficult to connect the dots and see how they support a central, significant claim. There are many individual experiments but insufficient details or motivation for them. For example, the authors introduced the concept of “refusal dilution” in section 5.4, but it is never elaborated or connected to section 6. Another example is the refusal direction experiments in section 5 - how are they related to your jailbreak method?

Also, the jailbreak experiments are conducted primarily on proprietary models, whereas the mechanistic analysis focuses solely on a single open-source model, Qwen3-13B. Is the mechanistic analysis trying to explain why your jailbreak method succeeds? If so, then the jailbreak evaluation and mechanistic analysis should be conducted on the same suite of reasoning models (or at least on overlapping sets of models).

**2. Missing details and unclear experimental design:**
- Table 2: It is unclear why the three baseline methods were chosen. Are they considered state-of-the-art?
- Using only GPT-4o-mini as the evaluator seems insufficient. The ASR could be inflated since GPT-4o-mini is not a highly capable model. Have you verified the alignment between GPT-4o-based evaluations and human annotations?
- Dataset selection: Why are jailbreak results reported on HarmBench, while refusal direction results are on JailbreakBench? This design choice is unclear.
- You mention a “Seduction” pipeline for automated jailbreaks, but it is not described anywhere. It is also unclear which auxiliary LLMs were used to generate the attacks.
- Tables 4 and 5: Why was DeepSeek Judge used for ablation and substring matching for addition? I understand that substring matching may be the simplest approach to detect refusals, but the evaluation should be standardized across experiments.

**3. Incorrect or missing citations:**
- Several important citations in the introduction are incorrect, including: HarmBench, Mousetrap, AutoRAN (L38-39). The author names are completely different from the actual works.
- Section 3 seems to be mainly prior work, see https://arxiv.org/abs/2502.12025.

**4. Writing issues:**
- Inconsistent usage of \citep and \citet in related work first paragraph
- L170: textbfNatural
- L234-243: the two paragraphs seem to be repeating each other
- Table captions should be above the tables
- L398: Missing figure link

**Questions:**

See weaknesses.

---

> ### Author Response · Authors · 2025-11-23
> **First Rebuttal~1**
>
> We sincerely thank the reviewer for the comprehensive and constructive feedback. We particularly appreciate your insightful comments regarding the **logical connection between our mechanistic analysis and jailbreak experiments** (Weakness 1), as well as the rigor of our **experimental design regarding evaluators and model selection** (Weakness 2). These points significantly helped us identify gaps in our narrative and strengthen the scientific quality of our work.
>
> # Response to Weakness #1: Overall
> > While this work presents a range of experiments, I find it difficult to connect the dots and see how they support a central, significant claim. There are many individual experiments but insufficient details or motivation for them.
>
> Thank you for your constructive criticism and for giving us a chance to sharpen our narrative.
> Here is the central story that connects our experiments:
> *   **Challenge to prevailing wisdom:** We contest the growing perception that reasoning models are inherently safer or that "more reasoning means more robustness." We demonstrate the exact opposite: the reasoning process itself introduces a new, critical attack surface.
> *   **Empirical Disproof (The Attack):** We introduce *CoT Hijacking* to prove that reasoning can undermine safety. By forcing the model to generate long, benign reasoning sequences before a final answer, we achieve state-of-the-art Attack Success Rates (e.g., 99% on Gemini 2.5 Pro), effectively turning the model's compute against its own safety guardrails.
> *   **Mechanistic Explanation (The Why):** We investigate the internal cause of this failure to "connect the dots" between the attack and the model's behavior. We establish that (a) reasoning models rely on a specific **refusal direction** in activation space to block harm, and (b) our attack succeeds because long benign contexts causally **dilute** this signal. As reasoning length increases, attention shifts away from the harmful instruction, causing the refusal feature to fade below the threshold required to trigger a rejection.
>
> > For example, the authors introduced the concept of “refusal dilution” in section 5.4, but it is never elaborated or connected to section 6. Another example is the refusal direction experiments in section 5 - how are they related to your jailbreak method?
>
> Thank you for the thoughtful consideration. *Refusal dilution* in section 5.4 is directly supported by the empirical evidence in **Figure 23** and **Appendix J**, and Section 6 further explains this phenomenon through mechanistic interpretability. We will revise Section 5.4 to more clearly articulate this connection and strengthen the motivation for introducing *refusal dilution*.
>
> Refusal direction experiments in section 5 demonstrate that refusal behavior **in CoT** can be mediated by one direction activation. This provides the conceptual and empirical foundation required for Section 6, where we analyze how CoT length interacts with this mechanism. In other words, Section 5 is intended as the prerequisite for Section 6. We will polish the exposition to ensure the transition reads naturally and the motivation is explicit in the updated version.
>
> > Also, the jailbreak experiments are conducted primarily on proprietary models, whereas the mechanistic analysis focuses solely on a single open-source model, Qwen3-13B. …… on the same suite of reasoning models (or at least on overlapping sets of models).
>
> Regarding model choices, proprietary reasoning models typically possess substantially stronger safety guardrails, which is why we evaluate jailbreak performance on them—this follows the standard practice established by prior baselines as well. To enrich the experimental scope, we have since conducted additional jailbreak evaluations on open-source reasoning models (DeepSeek-R1, Qwen3-Max, Kimi-K2-Thinking, and Seed-1.6-Thinking), all of which yield **100% ASR**. These results will be included in the updated version. For mechanistic analysis, we conducted experiments not only on **Qwen3-14B** but also on **GPT-OSS-20B** (in Appendix). We will explicitly highlight this in the revision.

---

> ### Author Response · Authors · 2025-11-23
> **First Rebuttal~2**
>
> # Response to Weakness #2: Missing details and unclear experimental design
>
> > Table 2: It is unclear why the three baseline methods were chosen. Are they considered state-of-the-art?
>
> We appreciate you raising this question regarding baseline selection. All three baseline methods are taken directly from the recent line of work on ‘jailbreaking large reasoning models’. Although these literatures are still extremely limited—the earliest representative work being **Mousetrap (Feb 2025)**—these baselines are considered the strongest available and report excellent performance in their original papers. We re-ran all baseline methods under the standardized JailbreakBench evaluation protocol and observed substantially lower ASR compared to their reported numbers.
>
> > Using only GPT-4o-mini as the evaluator seems insufficient... Have you verified the alignment between GPT-4o-based evaluations and human annotations?
>
> Thank you for scrutinizing our evaluation protocol. For the small preliminary experiments on **S1**, we used GPT-4o-mini as the judge because refusals elicited by direct malicious prompts are trivial to evaluate, which is consistant with human examination too. Our main jailbreak experiments, however, are judged by **Gemini 2.5 Pro** (as noted in the Appendix), which is a highly capable and good at long-context evaluator. Since our jailbreak method produces very long outputs, a strong long-context judge is essential. Using both robust judging protocols and strong evaluators consistently confirms the stability of our ASR results.
>
> > Dataset selection: Why are jailbreak results reported on HarmBench, while refusal direction results are on JailbreakBench? This design choice is unclear.
>
> We acknowledge that the use of different datasets might appear inconsistent. Regarding dataset choice: we report jailbreak results on **HarmBench** because it provides a compact set of high-quality harmful instructions, which helps control API cost while maintaining evaluation rigor. The *refusal direction* experiments, on the other hand, use **JailbreakBench** to remain consistent with prior work (Arditi et al., 2024).
>
> > You mention a “Seduction” pipeline for automated jailbreaks, but it is not described anywhere. It is also unclear which auxiliary LLMs were used to generate the attacks.
>
> We apologize for the lack of clarity regarding the pipeline details. The “Seduction” pipeline corresponds to **Figure 2**. Conceptually, it can be briefly explained as, the target model becomes absorbed in solving a complex puzzle—similar to being drawn into a momentum of continuous reasoning—before encountering the harmful payload. The auxiliary model used as an attacker is **Gemini 2.5 Pro**, as shown in Figure 2. We will provide a clearer description of this pipeline and the auxiliary model in the revised version.
>
> > Tables 4 and 5: Why was DeepSeek Judge used for ablation and substring matching for addition? ... the evaluation should be standardized across experiments.
>
> Thank you for pointing out the difference in evaluation metrics. For Tables 4 and 5: after manual inspection, we found that substring-matching was unreliable for the *refusal ablation* setting, so we adopted a lightweight LLM judge, which is sufficiently capable for this simple judging task. Conversely, substring-matching proved accurate for the *refusal addition* experiments after human verification. In summary, for each setting we selected the most efficient method that was empirically validated to be accurate, ensuring reliability across experiments without unnecessary complexity.
>
> # Response to Weakness #3: Incorrect or missing citations
>
> > Several important citations in the introduction are incorrect, including: HarmBench, Mousetrap, AutoRAN (L38-39). The author names are completely different from the actual works.
>
> Thank you for pointing this out. These citation errors likely stem from issues in our .bib file, and we will correct all mismatched references in the revised version.
>
> > Section 3 seems to be mainly prior work, see https://arxiv.org/abs/2502.12025.
>
> Regarding the similarity to the work you referenced, we acknowledge that there is somewhat overlap. However, key differences remain. Our *minimal* setting differs from their “less thinking” configuration in the specific content that is fixed, and our *extended* setting controls only the **initial tokens** of the CoT, which is distinct from their constant appending of “wait.” In our experiments, both the minimal and extended settings are designed in parallel, with each constraining only a targeted subset of CoT tokens rather than enforcing a constant intervention. We will add a proper citation to the referenced work and clarify these distinctions in the updated version.
>
> # Response to Weakness #4: Writing issues
>
> > Inconsistent usage of \citep and \citet in related work  ...... L398: Missing figure link
>
> Thank you for highlighting these writing issues. We will tackle all these problems in the updated version.

---

### Note · Program_Chairs · 2026-01-17
**Submission Desk Rejected by Program Chairs**

The following references in this submission do not refer to real documents and/or have major errors in bibliographic information:

 Chengzhi Chao et al. Mousetrap: An automated framework for jailbreaking and defending large language models. In International Conference on Learning Representations (ICLR), 2024a.
Xiangyu Chen et al. Autoran: Automated red teaming via reinforcement learning. In International Conference on Learning Representations (ICLR), 2023.